# Studying the Rationale of Fire Ant Sting Therapy Usage by the Tribal Natives of Bastar Revealed Ant Venom-Derived Peptides with Promising Anti-Malarial Activity

**DOI:** 10.3390/toxins14110789

**Published:** 2022-11-11

**Authors:** Jyoti Kumari, Raj Kumar Sah, Nazar Mohamed Mohaideen. S, Shakeel Ahmad, Soumya Pati, Shailja Singh

**Affiliations:** 1Department of Life Sciences, School of Natural Sciences, Shiv Nadar Institute of Eminence, Delhi 201314, India; 2Special Centre for Molecular Medicine, Jawaharlal Nehru University, New Delhi 110067, India; 3Institute of Physical Chemistry, Polish Academy of Sciences, Kasprzaka 44/52, 01-224 Warsaw, Poland

**Keywords:** ant venom, anti-malarial activity, *Plasmodium falciparum*, venom peptide, alkaloids, Bastar tribes

## Abstract

Prevailing drug resistance in malaria imposes the major roadblock for the existing interventions necessitating the timely need to search for alternative therapies. Ants in Solenopsis *spp*, termed ’Fire ants’, are well known for their aggressive behavior, which leads to the release of toxic venom. Notably, the tribal natives of the malaria-laden densely forested Bastar region, Chhattisgarh, India, use fire ant sting-based therapy to cure malaria-like high fever. Inspired by this, we have collected the fire ants from the forest of Bastar and extracted peptide and alkaloid fractions from ant venom using HPLC and analyzed them by LC/MS-based applications. Evaluation of the anti-malarial efficacy of these peptide fractions demonstrated a significant reduction in the growth of *Plasmodium falciparum* (*Pf* 3D7) in vitro, whereas the alkaloid fraction showed a negligible effect. in vitro hemolytic activity confirmed the venom peptide fraction to be non-hemolytic. Additionally, the venom peptide fraction is purely non-toxic to HepG2 cells. Anti-malarial efficiency of the same in *Plasmodium berghei ANKA* infected mice models showed a drastic reduction in parasitemia representing promising anti-malarial activity. Overall, our study has unraveled the scientific rationale underlying fire ant sting therapy used as a tribal naturotherapy for curing malaria-like fever, thus, introducing a way forward to develop nature-inspired anti-malarial chemotherapeutics.

## 1. Introduction

Malaria has persisted as one of the most deadly infectious diseases in the world. According to the World Health Organization (WHO) malaria report in 2020, 241 million cases have been detected so far with children under the age of 5 being known as the most affected in the endemic zones [1]. Malaria is caused by obligate intracellular parasites of the genus *Plasmodium* and is transmitted to humans by the *Anopheles* mosquito, which serves as a vector for dissemination. Of all the *Plasmodium species, Plasmodium falciparum* is known to cause the most serious and life-threatening infection in humans. Infection usually starts when a mosquito harboring the parasite feeds on the host blood and, in the process, releases sporozoites into the bloodstream [2]. These sporozoites infect hepatocytes in the liver where asexual reproduction produces large numbers of merozoites that further infect the erythrocytes [3]. Upon infection, these merozoites multiply rapidly, leading eventually to hemolysis. At this stage, a large number of merozoites enter healthy erythrocytes to start a new cycle of infection. The highest mortality rate in humans is reported to be at this stage of the infection [4]. Numerous reports indicated that resistance to the known drugs is progressively emerging against *Plasmodium* making it absolutely essential to look for newer and more effective drug targets [5,6].

Nature produces a wide range of bioactive molecules and, thus, offers chemists, biologists, and the pharmaceutical industry a plethora of information to explore new drug regimes [7,8]. Over the course of the 20th century, venom, a biochemical component for animal self-defense, isolated from multiple phyla of the animal kingdom, has been shown to have a wide array of medicinal properties including anti-bacterial, anti-cancer, and anti-inflammatory, etc. [9,10,11]. Most of the venomous animal species are found in phyla such as Chordata (reptiles, fishes, and amphibians), Arthropod (arachnids, Chelicerata, and insects), Mollusca (cone snails), Echinodermata (starfishes and sea urchins), and Cnidarian (sea anemones, jellyfish, and corals) [12,13]. Bee Venom extracted from class Insecta, *Apis mellifera, A. m. ligustica, A. m. scutellate,* have been effectively used as anti-inflammatory agents for managing diseases like rheumatoid arthritis, heart and skin diseases [9,10,14], and neurological impairments like Alzheimer’s and Parkinson’s disease [15]. Additionally, snake venoms have been shown to act as potential drug targets for anti-bacterial, anti-viral, anti-parasitic, and anti-fungal activities [16]. Venom-derived compounds like Cecropins (*Hyalophora cecropia)* [17] and scorpines (*Pandinus imperator)* have been shown to affect the malaria parasite’s life cycle at various stages [18].

It is noteworthy that ant venom was poorly investigated in contrast with the venoms of other species considering its taxonomic richness [19]. Among ants, predators are presented with a subfamily of Ponerinae [20]. Most of them take or capture prey using their toxic venom [21] which contains peptides [22,23]. The allergenic venoms of fire ants of the genus *Solenopsis* (Myrmicinae), *Pachycondyla* spp. (Ponerinae), or the Australian Myrmecia (Myrmeciinae) have especially been studied for their components [20,24]. Ant venoms have been found to contain an extraordinary diversity of toxins and other types of molecules including salts, sugars, formic acid, biogenic amines, alkaloids, free amino acids, hydrocarbons, peptides, and proteins [24,25,26]. The first reported case involving the therapeutic use of venoms from ants was against rheumatoid arthritis [27]. Recently, several studies reported pharmacological agents from ant venoms such as anti-inflammatory and immune-boosting drugs, etc. [27,28]. A recent study stated that *Pachycondila sennaarensis* venom has a substantial dose- and time-dependent anti-cancer effect on breast cancer cells [29]. Although the pharmacological properties of ant venom components have gained the attention of the scientific community, studies involving anti-malarial activity are still in their infancy.

Here we have explored the “fire ant sting therapy”, a nature-derived curative practice by native tribes of the Bastar District in Chhattisgarh, India, for treating malaria-like high fever. This study has deciphered the scientific rationale of this traditional tribal medicine and discovered the ant venom-derived peptide fraction with excellent anti-malarial efficacy both in vitro and in vivo.

## 2. Materials and Methods

### 2.1. Ant Collection and Venom Extraction

Fire Ants from the (Hymenoptera: Formicidae) *Solenopsis* species were brought from a village in Chhattisgarh in late August 2017 and maintained in 20 L plastic boxes painted with Fluon^®^ to prevent the ants from escaping [30]. Extractions were performed on the same day after the collection of the ants from the field using established protocol by [31]. Figure 1 shows the venom extraction procedure initiated with the collection of ants and then the ants were separated from the nest by slowly flooding the bucket with water. This process took several hours; thus, the venom extraction solution was prepared fresh. Once completely flooded, the ants form a raft at the surface of the water and were collected and placed in a 500 mL of extraction solution in a 1 L glass beaker containing water and hexane in a 1:5 ratio. The extraction mixture was clearly separated into two phases and the ants were completely immersed in the organic solvent. When the ants entered the organic solvent, they instinctively discharged their venom while sinking, perhaps due to their defensive nature, and then rapidly died. These two phases were easily separated into individual tubes using pipettes and separator glass funnels. The upper, organic phase contained venom alkaloids and cuticular hydrocarbons [32]. The lower, aqueous phase contained water-soluble proteins. These proteins were then extracted by lyophilizing this phase and resuspending it in the proper solution. Database: Insect base 2.0, Inscetbase ID: IBG_00711.

### 2.2. Protein Quantification by BCA Method

Working solutions of BCA were prepared by mixing 50 parts of reagent A (BCA, sodium carbonate, sodium bicarbonate, bicinchoninic acid, and sodium tartrate in 0.1 M sodium hydroxide) with 1 part of reagent B (CuSO4, 4%), as indicated by the manufacturer. Thereafter, 200 μL of BCA working solutions were pipetted onto the wells of a 96-well plate, and 25 μL of the samples’ peptides fraction in phosphate buffer saline (PBS) pH 7.4 was added, giving a BCA working solution with a sample ratio of 8:1. The plate was incubated for 30 min at 37 ℃ and then placed in the reader. Prior to reading, the plate was allowed to shake in a smooth motion for 30 s. The absorbance was read at 562 nm in a Varioskan LUX multimode microplate reader (Thermo Fisher Scientific, Waltham, MA, USA).

### 2.3. Extraction and Analysis of Venom Fractions by HPLC

The C18 reversed-phase high-performance liquid chromatography (RP-HPLC) was used for the analysis of the venom fraction and ant venoms were fractionated using an Xterra-C18 column (5 µm, 2.1 mm × 100 mm); Waters, Milford, MA, USA) with a gradient of solvent A (0.1% v/v TFA) and solvent B (ACN/0.1% v/v TFA). The percentage of solvent B was modified as follows: 0% for 5 min, 0–60% over 60 min, 60–90% over 10 min, and 90–0% over 15 min at a flow rate of 0.3 mL/min. The UV absorbance of the eluent at 215 nm. All analyses were performed on an HPLC system (Waters, Milford, MA, USA). The lyophilized venom peptide fraction was immediately filtered through an Amicon Ultracel 10 K centrifugal filter system (Millipore, Bad Schwalbach, Germany) with a 10,000 Da molecular mass cut-off to prevent venom proteolytic peptide cleavage.

### 2.4. Analysis of Venom-Derived Peptide Fractions in LC/MS

Extracted protein fraction samples were then subjected to liquid chromatography–mass spectrometry (LC/MS) analysis. We used a Waters Acquity H-Class UPLC system (Waters, Milford, MA, United States). Chromatographic separation was achieved on an Acquity BEH C18, 1.7 μm, 75 mm × 2.1 mm column (Waters, Manchester, United Kingdom). The mobile phase consisted of 0.1% formic acid solvent (A) and acetonitrile (B). The initial gradient condition constituted 90% A and 10% B, 80 and 20% for 2 min, 50 and 50% for 3 min, 20 and 80% for 1 min, 10 and 90% for 2 min, and then linearly changed to 34% B over 8 min, and turned back to the initial condition of 90% A and 10% B, and then washed up to 15 min. The column temperature was adjusted to 35 °C. The flow rate was 0.3 mL/min, and the injection volume was 5 μL. Mass spectrometry was performed in a negative electrospray mode using a high-resolution mass spectrometer SYNAPT G2 S HDMS (Waters, Manchester, United Kingdom) with a TOF-detector with a linear dynamic range of at least 5,000:1. The desolvation gas (45 °C, 647.0 L/h) and the nebulizer gas (6.0 bar) were nitrogen. The cone gas had a flow of 52 L/h. The capillary voltage was 2.52 kV and the source temperature was 90 °C. The analyzer mode was set at “resolution” and the dynamic range at “extended.” The mass spectra were acquired over the range of 550–3000 Da with a spectral acquisition rate of 0.1 s per spectrum.

### 2.5. Cytotoxic Assay

The HepG2 cells, derived from the human liver, were obtained from the American Type Culture Collection (ATCC, Manassas, VA, USA). They were cultured in Dulbecco Modified Eagles Medium (DMEM) with L-glutamine (Gibco, Grand Island, St. Louis, NY, USA) enriched with 10% heat-inactivated FBS (Fetal Bovine Serum, Gibco, Grand Island, NY, USA) and 1% penicillin-streptomycin (Gibco, Grand Island, NY, USA) at 37 °C. MTT 3-(4,5-dimethylthiazol-2-yl)-2,5-diphenyltetrazolium bromide cell proliferation assay was performed according to the standard protocols (ATCC^®^ 30-1010K). HepG2 cells were trypsinized (Trypsin-EDTA, Gibco, Grand Island, NY, USA), counted, and seeded at a density of 10,000 cells per well in complete DMEM in a 96-well plate and incubated at 37 °C for 20–24 h. The media was then discarded carefully and fresh complete DMEM was added; the test wells were treated with 25 ug/mL of the peptide-like fractions, whereas the untreated well was used as a control, further, the treatment was given for 20–24 h. After treatment, 10 µL of MTT (Sigma, Saint Louis, MO, USA)) at a concentration of 5 mg/mL in phosphate buffer saline (PBS) was added to both untreated and treated wells and incubated for 2–3 h until a purple precipitate was visible. The formazan crystals were seen and dissolved in 100 μL DMSO (Dimethyl sulpoxide) and the plate was shaker incubated for 15 min. Absorbance was taken at 570 nm and a reference wavelength of 630 nm was recorded using a Microplate Reader (ThermoFishcer, Varioskan LUX, MA, USA).

### 2.6. Hemolytic Activity

Hemolytic activity was calculated by incubating human RBC (O positive blood group) suspensions with serial ant venom peptide dilutions. RBCs have been centrifuged to PBS for 3 min at 3,000 g multiple times until the optical density (OD) of the supernatant reached the control OD (PBS only). Human erythrocytes (2% haematocrit) in suspension were treated with 1% Triton X-100 (positive control) and with PBS containing 1, 5, 10, 20, and 40 μg/mL of ant venom peptides, and then incubated in a shaking water bath at 37 °C for 3 h. The samples were then centrifuged at 3000 g for 5 min, the supernatant was isolated, and the absorption was measured at 540 nm as an index of released haemoglobin. The relative optical density was calculated and represented as a percentage of haemolysis compared to the suspension treated with 1% of Triton X-100.

### 2.7. In Vitro Antiplasmodial Activity 

Frozen aliquots of *P. falciparum* field isolates and laboratory strains were obtained from the Malaria Research and Reference Reagent Resource Centre (MR4), Manassas, VA, USA. Parasites were thawed and cultured using standard protocols [33]. A 48-h in vitro growth inhibition assay [34] was used to test the anti-malarial activity of ant peptide and alkaloid fractions. The chloroquine-sensitive strain *Plasmodium falciparum* 3D7 was cultured using standard protocol. *P.falciparum* line 3D7 was cultured in RPMI-1640 (2 mM L-glutamine, 25 mM HEPES, 2 g L^−1^ NaHCO_3_, 27.2 mg L^−1^ hypoxanthine and 0.5% Albumax II, pH 7.4) medium with 2% hematocrit (O positive Human erythrocyes) and incubated in a humidified atmosphere with 5% O_2_, 5% CO_2_, and 90% Nitrogen at 37 °C. The parasites were synchronized by treatment with 5% sorbitol when the ring stage was observed under a microscope. Thereafter, 200 µL of the culture with a parasitaemia of 0.5% ring parasites and a hematocrit of 2% was added into a 96-well plate in triplicate with ant peptide and alkaloid concentrations (1.442, 2.884, 4.326, 5.762, 7.21, 8.662) µg and (0, 2, 6, 4, 8, 10) µL, respectively. The anti-malarial efficiency of the ant peptide was estimated 48 h post-treatment and the parasite re-invaded RBCs at the ring stage. The live parasites of thin blood films stained with Giemsa were counted under a microscope. Total parastemia of treated cultures was compared with the control (only Infected RBCs) with counted 3000–4000 RBCs. The percentage of parasite growth suppression (PGS) was estimated using the following formula:$$\mathrm{Parastemia}\%=\frac{Control-Treated}{Control}\times 100\%$$

#### SYBR Green Assay

To validate the growth inhibition of *P. falciparum* 3D7, the 96 well plates treated with both alkaloid and peptide fractions were centrifuged at 2000 rpm for 5 min. Following the centrifugation, the supernatant was discarded and then SYBR green (100 µL) was added to each well and plates were incubated for 2–3 h. The growth inhibition rate was measured using a Varioskan™ LUX multimode microplate reader (ThermoFisher, Varioskan LUX, Waltham, MA, USA).

### 2.8. Animal Handling and Parasite Inoculation

Animal studies were performed in accordance with the guidelines of the Institutional Animal Ethics Committee (IEAC) of Jawaharlal Nehru University (JNU), Delhi, and the Committee for Control and Supervision of Experiments on Animals (CPCSEA). The experiments performed and the use of laboratory animals were approved and were in strict accordance with the ethical guidelines approved by the animal ethics committee IAEC-JNU. The mice obtained from Central Laboratory Animal Resources, Jawaharlal Nehru University, Delhi, were housed under standard conditions of food, temperature (25 °C ± 3), relative humidity (55 ± 10%), and illumination (12 h light/dark cycles) throughout the experiment. The chloroquine-sensitive *P. berghei* ANKA strain was obtained from the Ethiopian Public Health Institute and maintained at an animal house facility. For parasite maintenance, serial passages of blood from infected mice to non-infected ones was performed. A donor mouse with a parasitemia of approximately 30% was sacrificed and blood was collected in a microcentrifuge tube containing citrate phosphate dextrose (CPD) as an anticoagulant. The blood was then diluted with 0.9% normal saline. Each mouse used in the experiment was inoculated intraperitoneally with 0.2 mL of 1 × 10^7^
*P. berghei-infected* RBCs [35].

### 2.9. In Vivo Anti-Malarial Activities

The four-day suppressive test was used to measure the anti-malarial activity of ant venom protein fraction against *P. berghei-infected* Balb/c mice. In brief, infected mice were randomly divided into three groups of 4 each by weight. Mice were infected by intraperitoneal (i.p) inoculation of 1.0 × 10^7^ infected erythrocytes from donor mice on the first day (day 0) of the experiment. Peptide treatment 1 mg/kg body weight was administrated within 1 h post-inoculation of mice with the parasite (day 0) in a dose volume of 0.2 mL. The positive control group received 10 mg/kg body weight quinine orally per day, while the negative control group animals were administered 200 µL of phosphate buffer saline (PBS). Mice were dosed daily by i.p injection for 4 consecutive days. For all of these days, Giemsa-stained thin blood smears were prepared from the tail of each animal and stained with 10% Giemsa for 15 min and examined under a microscope at 100X. The percentage of parasitemia was determined by counting parasitized RBCs on at least 3000 cells. The survival rate was monitored daily in all groups for 20 days post-inoculation.

### 2.10. Statistical Analysis

Statistical analysis was conducted with GraphPad Prism 8.0 (GraphPad Software Inc., GraphPad Prism 8.0.1.244, San Diego, CA, USA, 2018). Data were analysed statistically using one-way ANOVA and two-tailed Student’s t-test to identify the differences between the treated group and the controls. All data are presented as means ± SEM. A value of *p* < 0.05 was considered significant.

## 3. Results

### 3.1. Quantification of Peptide-like Fractions by BCA Method

To check the concentration of peptide-like fraction, we performed BCA. The nine-point calibration curve was plotted in duplicate from the BSA included in the kit with the concentration range from 0 to 2000 µg/mL. Dilutions of 10 × for the ant venom peptide fraction were used to measure peptide concentration in the venom. The concentration of ant peptide was estimated as 0.443 µg/µL by BCA.

### 3.2. Effect of the Ant Venom Peptide Fraction against In Vitro Growth of Plasmodium falciparum

To evaluate the effect of ant venom fraction containing protein/peptides or alkaloids on parasite growth of the asexual blood-stage of the *P. falciparum* parasite, we treated the parasite at ring stage with an increasing concentration of protein and peptide fraction (1.442–8.662 µg/mL) and allowed them to grow for one cycle. The growth rate of the parasite was evaluated using SYBR Green assay and Giemsa counting. The results demonstrated that peptide fraction strongly inhibited parasite growth in a dose-dependent manner (1.442–8.662 µg/mL). The IC_50_ was calculated and found to be 6.03 μg/mL (Figure 2A) while the alkaloid fractions showed a trivial effect on growth inhibition (Figure 2C). The representative Giemsa-stained blood smears also demonstrated a halt in the parasite growth, which could account for the formation of a “pyknotic body” after 48 h of the treatment with ant peptide fractions as compared to the control (untreated) (Figure 2B), while the Giemsa images for alkaloid treatment showed a negligible effect (Figure 2C,D). 

To determine the effect of venom-derived components on RBC hemolysis, we have treated human RBCs with different concentrations of peptide and alkaloid fractions (1 to 40 µg/mL) as mentioned in the method section. The indication of hemolysis is represented by the destruction of RBCs followed by the escape of hemoglobin from the same. The results indicated that even at the higher concentration (40 µg/mL) both the peptide and the alkaloid fractions had insignificant hemolytic activity, as compared to the positive control 1% trionX-100 Figure 2E. These findings suggest that the venom components have no effect on hemolysis.

Since our principal aim is to study the fire ant venom for the development of novel anti-malarial compounds, it is also necessary to check whether these extracts exhibit cytotoxicity in human cells. Thus, we have elucidated their cytotoxic effect on the HepG2 cell line. The results represented that the survival efficiency of HepG2 cells after exposure to 1, 5, 10, 20, and 40 µg/mL of the venom was found to be 98.45, 96.67, 98.03, and 97.32 percent, respectively (Figure 2F). This data strongly indicated that the venom peptide fraction is purely non-toxic in nature.

### 3.3. In Vivo Anti-Plasmodial Activity of the Peptide Fraction 

To determine the anti-malarial activity of ant venom peptide fraction, a *Plasmodium berghei (P. berghei)* infected mouse model has been used. To perform the classical four-day suppressive treatment in vivo, the intraperitoneal (i.p) mode of administration was used. A schematic representation of the four-day suppression test has been described in Figure 3A. While chloroquine treatment eliminated the infection in the positive control mice after two days of administration (with a dose of 25 mg/kg body weight), parasitemia rates were found to be increased in the vehicle mice as expected. Interestingly, the peptide fraction significantly reduced parasitemia (19.6 ± 3.14%) and showed extended dose-dependent survival in treated mice as compared to the vehicle group that showed high parasitemia (28.06 ± 0.96%). Moreover, peptide-treated mice lived longer than mice in the corresponding negative test classes. Precisely, the peptide extract exhibited a higher effect on the care groups’ mean survival period relative to the untreated control.

### 3.4. Analysis of Venom Protein Fraction Using HPLC and LC/MS

The HPLC analysis of venomous protein fractions revealed a distinctive peak at 215 nm that correspond to the presence of peptides in the venom protein fraction. The LC–ESI-MS analysis of the crude venom and active fraction of proteins showed an average molecular weight ranging from 615 to 2500 Da (Figure 4) which suggested the presence of peptides in fraction extracted from venom. 

**Database search:** Genetic information of *Solenopsis* species implemented by using NCBI database (http://www.ncbi.nlm.-nih.gov/, accessed on 20 October 2022) and GenBank assembly accession: GCA_000188075.2(https://www.ncbi.nlm.nih.gov/Taxonomy/Browser/wwwtax.cgiid=13686, accessed on 20 October 2022).**Strain information**: Laboratory strains *Pf* 3D7 (chloroquine-sensitive cell line) were obtained from the Malaria Parasite Bank at Malaria Research and Reference Reagent Resource Centre (MR4), Manassas, VA, USA.Both the databases accessed by 20 October 2022.

## 4. Discussion

The daunting task of eliminating the malaria burden across the globe is to discover novel anti-malaria that would aid in combating the existing roadblocks linked to available malaria therapeutics; namely, drug resistance and toxic side effects, etc. Nature-inspired studies have always untangled multiple scientific mysteries leading to new-age drug discovery. Supportive literature mining suggested that venoms from spiders, snakes, and bees, etc., harbour repertoires of bioactive molecules implicated in therapeutic implications involved in cancer, inflammation and pain response, microbial infections, etc. [36,37].

Moreover, accumulating studies have also identified important biomolecules from various venomous animals with promising pharmacological applications. Especially, the peptides and proteins from the venoms of snakes have shown excellent anticancer properties, as shown by prevention of cancer growth, inhibition of cancer invasiveness and cell cycle progression, and, finally, cancer cell apoptosis [36]. Similar studies have also identified the venom-extracted peptide from scorpions (*Mesobuthus martensii)* with prominent antibacterial properties [38]. Notably, Marcin-18, one of the antimicrobial peptides (AMPs) from scorpion venom (*Mesobuthus martensii*.), has gained a lot of interest for its beneficial role in combating antibiotic resistance [38]. Venom from scorpions has represented a wide variety of bioactive peptides and proteins. Due to its potent anti-inflammatory properties, scorpion venom peptides have been frequently used in the treatment of cardiovascular diseases. Besides this, it can also act as a potent anti-thrombotic and analgesic agent [39].

Plausibly, reviewing literature for venom-derived therapies also disclosed “Apitherapy” as one of the widely applied alternative medicinal approaches derived from honeybee stings against neurological diseases including ALS (Amyotrophic lateral sclerosis), Parkinson’s, and Alzheimer’s disease [40]. In addition, studies from bee venom therapy also identified many bioactive molecules such as peptides like melittin, apamin, mast cell degranulating (MCD) peptide, adolapin, and enzymes namely phospholipase A2 (PLA2) and hyaluronidase with therapeutic applications [41]. Ants and their medicinal importance draw attention to the pharmacological therapeutics of the 20th century. 

To emphasize, venoms from Arthropods, especially ants that belong to a large phylum, are rich in bioactive molecules with possible pharmacological importance, which is yet to be explored in detail. One of the known ant venoms which exert anticancer activity, namely, “*Samsum* ant venom” (SAV), contains a diverse collection of organic substances with exceptional pharmacological qualities that have demonstrated promising anti-neoplastic activity in various cancer cell lines [42].

In line with this information, our group also searched for nature-driven clues from the tribal-dominated area of the Bastar region, one of the heavy malaria-affected regions of India. We explored the “fire ant sting-based therapy” that the tribal natives of this region use to treat people suffering from malaria-like high fever. To further explore the malaria-suppressive and curative essence of the venom extracted from these fire ants, we have performed crude venom extraction using an established method [31]. Next, we identified alkaloids and peptide fractions using HPLC and LC/MS-based analyses. Though screening for hemolytic activity showed both fractions to be non-hemolytic even at higher concentrations like 40 µg/mL, only the venom-derived bioactive peptide fraction harbours strong anti-malarial activity with IC 50 of 6.03 µg/mL. This was also clear from the intracellular growth inhibition of P. falciparum in vitro following treatment with the venom peptide fraction (Figure 2), while the alkaloid fraction had a trivial effect on the intracellular growth of parasites. Additionally, the venom peptide fraction was also found to be purely non-toxic to HepG2 cells even at higher concentrations (Figure 2). Based on this analysis, it is assumed that peptide fraction would also have a negligible effect on hepatocytes during the early stages of malaria infection in the vertebrate host [3], thus, it can be proposed as a potent candidate for anti-malarial development. Furthermore, evaluation of the anti-malarial activity of the peptide fraction in the P. berghei ANKA murine model showed a significant reduction in parasitemia and showed enhanced survival efficiency, as compared to the control (Figure 3). Treatment with the venom peptide fraction significantly reduced parasite growth in mice models both in vitro and in vivo (Figure 5).

Furthermore, based on our present observations, we plan to identify the peptide-like fractions and their potent targets to initiate their clinical implications in the future. We will proceed with the fractionation of peptides from venom which will be further sequenced using automated Edman degradation, with obtained amino acid sequence information, and peptides will be synthesized and tested for anti-malarial activity. This will enormously help in the development of peptide-based anti-malarial chemotherapeutics. 

## 5. Conclusions

This is the first study where we have solved a scientific puzzle chosen from a nature-guided cue used as a tribal remedy for the treatment of malaria-like high fever. Our data have laid a strong scientific groundwork for exploring the hidden molecular cues underlying this traditional tribal naturotherapy in practice within the dense forest region of Bastar, a malaria endemic zone of India. Future study would be necessary to understand the whole ant venom components since several precursors comprise hypothetical and predicted toxins/polypeptides with unknown functions. Overall, our findings have unraveled ant venom as an excellent resource for developing alternative anti-malarial pharmacotherapeutics.

## Figures and Tables

**Figure 1 toxins-14-00789-f001:**
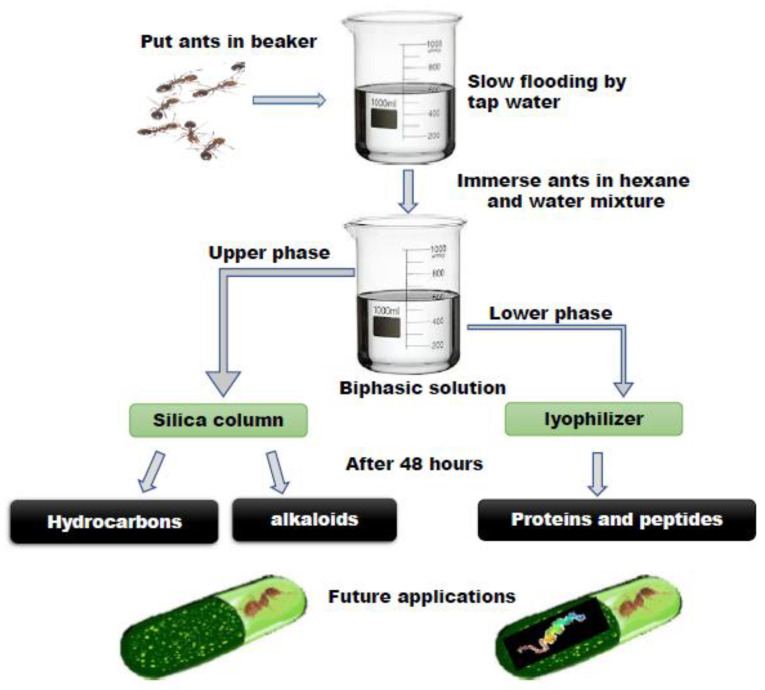
Schematic representation of the step-by-step isolation of the venom extraction.

**Figure 2 toxins-14-00789-f002:**
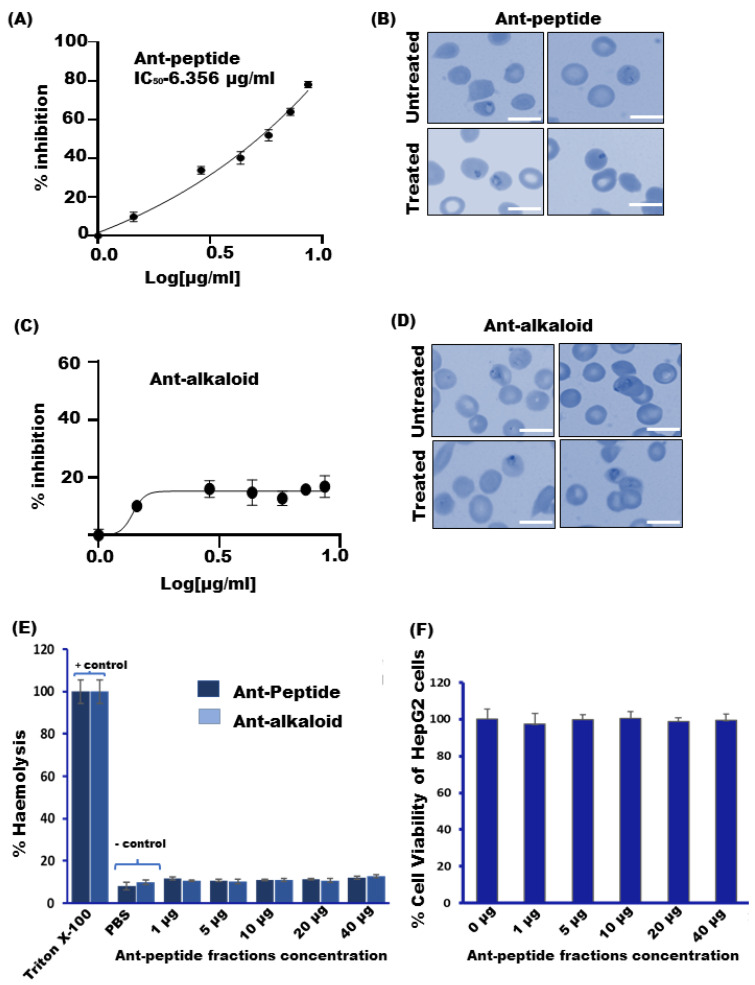
The venom peptide fraction showed parasite growth inhibition and the negligible effect on haemolysis and cell viability in vitro. (**A**) Infected RBCs were treated with varying concentrations of venom peptide (1.4–8.6 μg/mL) with or without for one intra-erythrocytic cycle (48 h). Untreated parasite culture served as a control. The parasitemia was scored on Giemsa-stained smears. The venom peptide-mediated growth inhibition of *P. falciparum* was found to be significant. (**B**) Alkaloid fraction treatment with concentration (1.4–8.6 μg/mL) showed minimal effect on growth inhibition. (**C**,**D**) Visualization of *P. falciparum* growth inhibition following peptide and alkaloid treatment was depicted by light microscopic images of Giemsa-stained slides. Human RBCs (2% haematocrit) in suspension were treated with 1% Triton X-100 (positive control), and with PBS containing 1, 5, 10, 20, and 40 μg/mL of ant venom peptides, and then Absorbance was read at 540 nm (**E**) The data has been represented as percentage lysis (mean of hemolysis ± SD). (**F**) Cytotoxicity effects of venom peptide on HepG-2 cell lines using MTT assay. HepG-2 cells were cultured in absence (control) and presence of different concentrations of the venom peptide fractions (1, 5, 10, 20, and 40 μg /mL) for 48 h. Cell viability Data was represented as means ± SD of two independent experiments (*n* = 2) with three biological repeats. Statistical analyses were performed with a Student’s *t*-test for unequal variances. *P*-value (Student’s t-test for unequal variances) <0.05 and <0.01 were considered significant. Scale bars = 10 μm.

**Figure 3 toxins-14-00789-f003:**
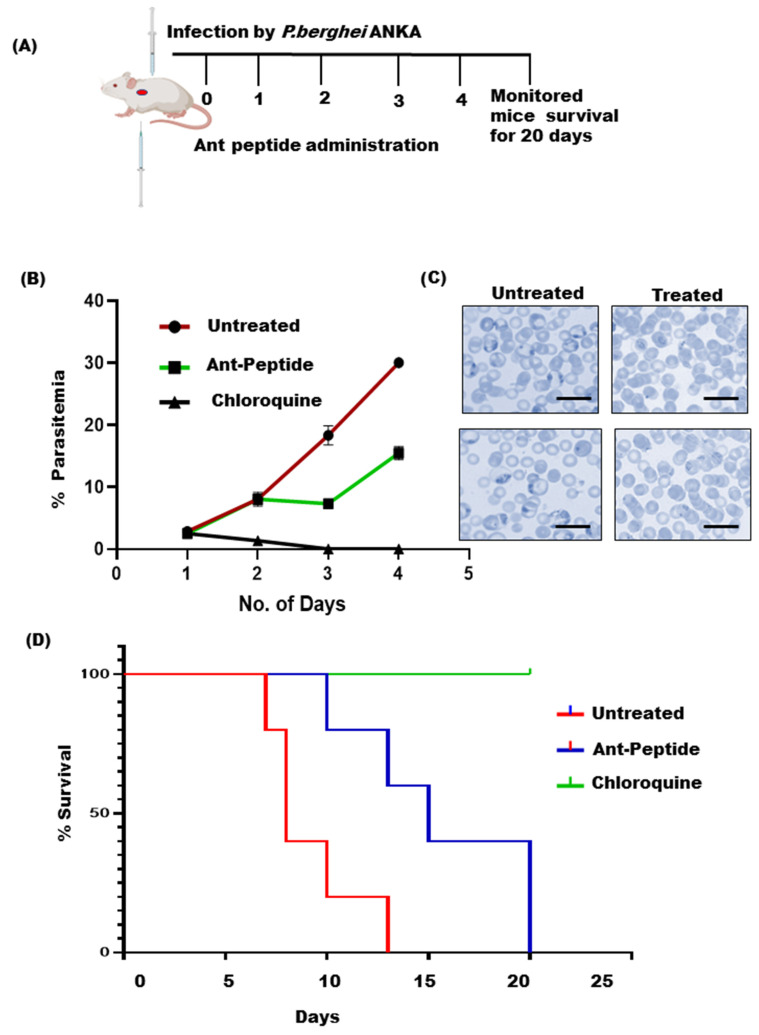
The in-vivo effect of the venom peptide fraction was studied in the *P. berghei-infected* mouse model. (**A**) Schematic representation of the four-day suppression treatment. (**B**) Groups of BALB/c mice (n = 4 per group) were inoculated with 1 × 107 parasitized erythrocytes of *P. berghei* ANKA by IP injection. After 3 h of infection, the venom peptide dissolved in PBS was injected intraperitoneally at 1 mg/kg in mice for 4 consecutive days. The infected mice without any treatment served as an untreated group. A dosage of 25 mg/kg body weight of chloroquine was used as a positive control. Parasitemia was monitored every day by drawing a smear from the tail blood of each group of mice. The graph represented the growth rate of *Plasmodium* upon treatment for 4 consecutive days. (**C**) Visualization of growth inhibition following peptide treatment was depicted by light microscopic images of Giemsa-stained slides. (**D**) Survival rates were determined, and peptide-treated mice showed more survival days as compared to the untreated control. Experiments were performed with (n = 4 for each group). Scale bars = 10 μm.

**Figure 4 toxins-14-00789-f004:**
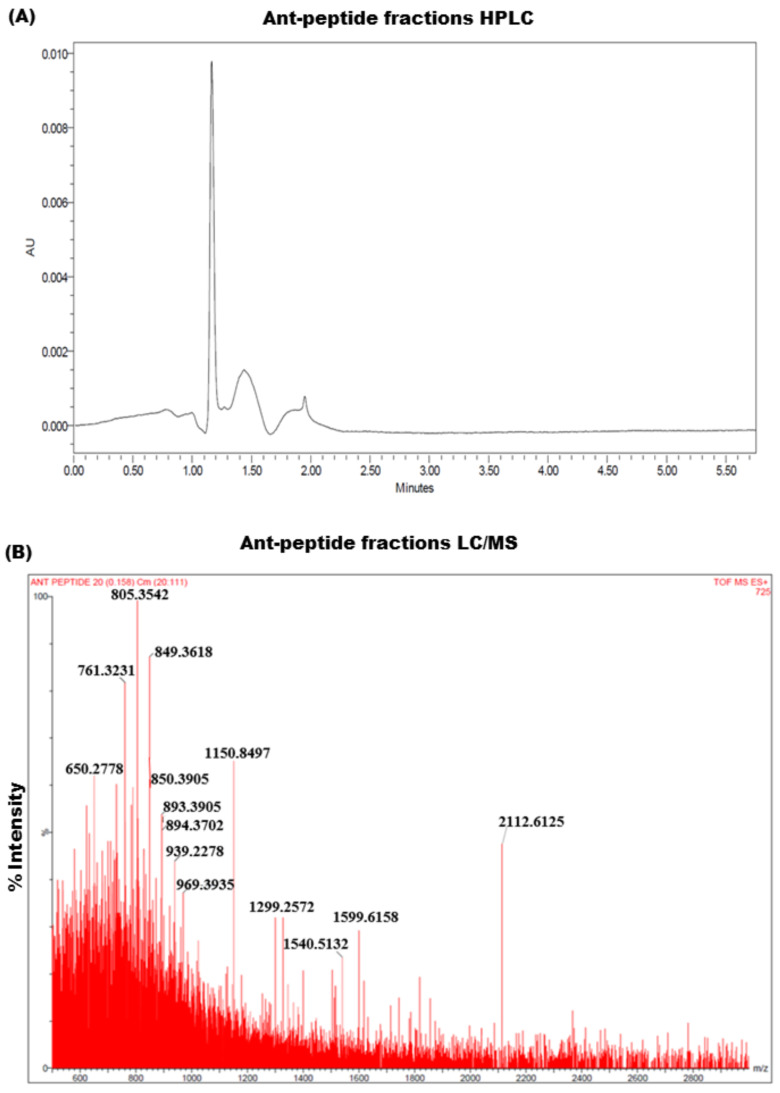
(**A**) Crude venom of the *Solenopsis* species (fire ant) was analysed by HPLC on a C18 reversed-phase column with gradient elution using water and acetonitrile supplemented with 0.1% formic acid as mobile phases. (**B**) The LC/MS mass spectra of the water-soluble peptide fraction. The masses of peptides extracted from the ant venom were identified in the corresponding peptide fraction.

**Figure 5 toxins-14-00789-f005:**
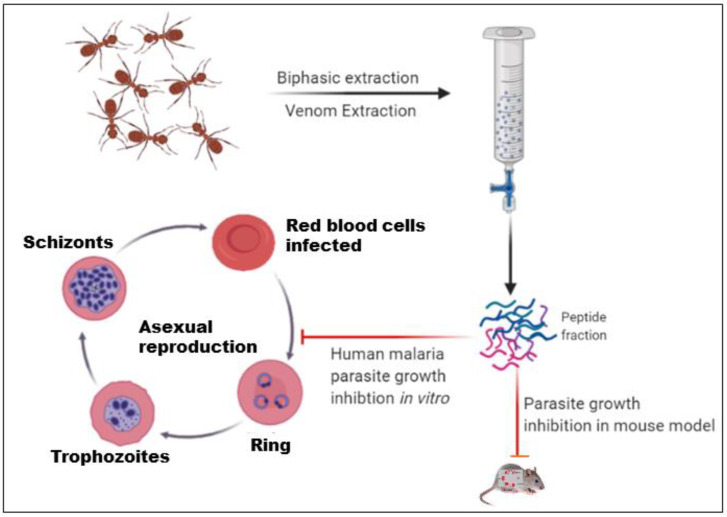
In the proposed model, the life cycle of the blood-stage parasite is shown. The venom peptide fraction treatment led to a significant reduction in parasite growth in both in vitro and in vivo mice models.

## Data Availability

Not applicable.

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
