# Peer review of "Studying the Rationale of Fire Ant Sting Therapy Usage by the Tribal Natives of Bastar Revealed Ant Venom-Derived Peptides with Promising Anti-Malarial Activity"

_toxins, 2022, doi:10.3390/toxins14110789_

Round 1

Reviewer 1 Report

The manuscript presents original data that are useful for the development of new antimalarials. The results show the potential of venom-derived molecules. Despite this, several aspects need to be reviewed. The methodology section needs more detailed descriptions. The sequence of methodology and results must establish a logical and clear relationship. The results should be discussed more clearly.  In this context, I do not recommend the manuscript for publication at the current status.

1. Some sentences need to be supported by references. For examples, lines 43-48.

2. Line 83. protocol by... Please include authors. 

3. Lines 108-109. This is a result and should be include in the appropiate section.

4. The chromatogram (Figure 4) does not show all the analysis time described in the methodology.

5. Which molecular weight marker was used in electrophoresis? Include in the methodology.

6. Have controls been added to the MTS assay? Was venom incubated with cell-free MTT? Can venom with MTT?

7. What blood type was used in the hemolytic assay?

8. Line 190. Scientific name must be written in italics.

9. Figure 1 was not described in the manuscript.

10. Figures must be presented in the order they are described in the manuscript. Revise lines 239-244.

11.  Figure 2 C. Y axis must end in a number.

12. µg is the correct symbol for the metric measurement microgram. Please revise figure 2C.

13. X-axis units must be identified.

14. Figure 5 was not described in the manuscript.

15. The discussion must be enriched.

16. The authors mention alkaloids, but do not present evidence of their identification.

17. Despite being described in the methodology, no gel electrophoresis was shown.

18. How do the authors confirm that the fractions only contain peptides or alkaloids?

Author Response

Response to reviewer’s comments

Reviewer #1: The manuscript presents original data that are useful for the development of new antimalarials. The results show the potential of venom-derived molecules. Despite this, several aspects need to be reviewed. The methodology section needs more detailed descriptions. The sequence of methodology and results must establish a logical and clear relationship. The results should be discussed more clearly. In this context, I do not recommend the manuscript for publication at the current status.

Comment1: - Some sentences need to be supported by references. For examples, lines 43-48.

Response: -Reference for lines 43-48, has now been added in the revised manuscript.

Comment 2: -Line 83. protocol by... Please include authors. 

Response: -As suggested by the reviewer for line 83 reference for the protocol has now been added to the revised manuscript.

Comment 3: -Lines 108-109. This is a result and should be included in the appropriate section.

Response: - As suggested by the reviewer, for lines 108-109, we have included this in the result section.

Comment 4: -The chromatogram (Figure 4) does not show all the analysis time described in the methodology.

Response: - The chromatogram in Figure 4, the HPLC chromatography, and the time analysis described in the methodology are for the solvent B flow rate. The chromatogram shown in the figure is for the eluent (peptide-like fraction) run with respect to the retention time is 0 to 5.50 minutes and its absorbance was monitored at 215 nm by a U.V detector.

Comment 5: -Which molecular weight marker was used in electrophoresis? Include in the methodology.

Response: -

The peptide fraction of ant venom was analyzed by LC/MS. Although we have the initial quality check of the peptide fraction. The SDS-PAGE did not resolve better. We crudely analyzed the peptide on gel electrophoresis because of its lower size it was a blotch, now we also removed this from our manuscript. As we have shown the data by HPLC and LC/MS. Since the above information was removed, thus the information regarding molecular weight marker is no longer needed here.

Comment 6: -6. Have controls been added to the MTS assay? Was venom incubated with cell-free MTT? Can venom with MTT?

Response: - Venom was directly used for the extraction of components as described by Fox et al 2012. So prior to extraction, no MTT was performed. Respected untreated controls were used for MTT analysis performed for ant-peptide treatment in vitro.

Comment 7: - What blood type was used in the hemolytic assay?

Response: -O+ (O positive) Red blood cells were used in the hemolytic assay.

Comment 8: - Line 190. Scientific name must be written in italics.

Response: -As suggested by the reviewer in line 190 the scientific name is now written in italics.

Comment 9: - Figure 1 was not described in the manuscript.

Response: -Thanks for your correction. We have inserted Figure 1 was now has been described in the methodology section of our revised manuscript.

Comment 10: - Figures must be presented in the order they are described in the manuscript. Revise lines 239-244.

Response: -As suggested, lines 239-244 have been revised with the correct citation of figures.

Comment 11: -.  Figure 2 C. Y axis must end in a number.

Response: - As suggested, Figure 2C has been corrected.

Comment 12: - µg is the correct symbol for the metric measurement microgram. Please revise figure 2C.

Response: - Sorry for the typological error. We have corrected this in figure 2C.

Comment 13: - X-axis units must be identified.

Response: -As suggested we have now clearly shown X-axis units in the revised manuscript.

Comment 14: - Figure 5 was not described in the manuscript.

Response: - As suggested we have included Figure 5 in the revised manuscript.

Comment 15: -The discussion must be enriched.

Response: - Thanks for the suggestion. We have updated the discussion as suggested.

Comment 16: - The authors mention alkaloids, but do not present evidence of their identification.

Response: -As suggested, we have added the references demonstrating the methods for alkaloid extraction and identification from Solenopsis invicta by Fox et al, 2012 that we have also used in the methodology for our manuscript.

Comment 17: - Despite being described in the methodology, no gel electrophoresis was shown.

Response: -The peptide fraction of ant venom was analyzed by LC/MS. Although we have the initial quality check of the peptide fraction. The SDS-PAGE did not resolve better. We crudely analyzed the peptide on gel electrophoresis because of its lower size it was a blotch, now we also removed this from our manuscript. As we have shown the data by HPLC and LC/MS.

Comment 18: - How do the authors confirm that the fractions only contain peptides or alkaloids?

Response: -We have used an established protocol by Fox et al,2012, which discussed a clear biphasic extraction system that can lead to the enrichment of both alkaloids and peptide fractions using hexane and water from Solenopsis Invicta. Further, the separated upper and lower phases were then processed for the isolation of alkaloid and peptide fractions using hexane-acetone and lyophilizer separately. This method was also supported by Li Chen et al. 2009, who also used the same protocol using silica gel chromatography to extract alkaloids from fire ant venom.

Reviewer 2 Report

This paper describes antimalarial activity of the peptide fraction from the venom of fire ants. This study was conducted based on the “fire ant sting therapy” used by the tribal natives, and proved the “peptide fraction” shows antimalarial activity. The results are quite interesting and useful for the future drug discovery and clinical application. Accordingly, this paper deserves to be published in this journal, but some points may be reconsidered and revised as follows:

1.     Are antimalarial components really venom-derived peptides? It is not chemically demonstrated. Actually, the water-soluble fraction is antimalarial, but the alkaloid fraction is not. The water-soluble fraction may contain not only peptides but also proteins, carbohydrates etc, although the mass spectrum indicates the major components are peptides. So, at this moment, it is not decisive that venom-derived peptides are responsible for antimalarial activity. To demonstrate peptides are antimalarial components, purification and identification of the peptides showing antimalarial activity are needed.

2.     Figure 4 A): for HPLC profile, it is better to show TIC (total ion chromatography) by mass spectrometry because it is much more sensitive and better resolution over the UV detection.

3.     Figure 4 B): this must be mass spectrum of  “water-soluble fraction” instead of  “peptide fraction”. As mentioned above, it is not demonstrated that all the components in this fraction are peptides.

Author Response

Response to reviewer’s comments

Reviewer #2

This paper describes antimalarial activity of the peptide fraction from the venom of fire ants. This study was conducted based on the “fire ant sting therapy” used by the tribal natives, and proved the “peptide fraction” shows antimalarial activity. The results are quite interesting and useful for the future drug discovery and clinical application. Accordingly, this paper deserves to be published in this journal, but some points may be reconsidered and revised as follows:

Comment 1: -    Are antimalarial components really venom-derived peptides? It is not chemically demonstrated. Actually, the water-soluble fraction is antimalarial, but the alkaloid fraction is not. The water-soluble fraction may contain not only peptides but also proteins, carbohydrates etc, although the mass spectrum indicates the major components are peptides. So, at this moment, it is not decisive that venom-derived peptides are responsible for antimalarial activity. To demonstrate peptides are antimalarial components, purification and identification of the peptides showing antimalarial activity are needed.

Response: -Thank you for your suggestion. We have used the established protocol by Fox et. al., 2012 for the isolation of peptide fraction from the aqueous phase, and the spectra data from LC/MS analysis clearly represented high abundance of peptides of lower size suggesting enrichment of the majority of peptides. Treatment with this peptide fraction showed an IC50 6.03 μg/ml in vitro and ~70% survival efficacy in vivo, representing strong antimalarial efficacy.

Comment 2: - Figure 4 A): for HPLC profile, it is better to show TIC (total ion chromatography) by mass spectrometry because it is much more sensitive and better resolution over the UV detection.

Response:   For the RP-HPLC, the UV absorbance of the eluent was monitored at 215 nm. And the resultant spectra peak was detected within 5.50 minutes as represented in Figure 4 A.

Comment 3: -    Figure 4 B): this must be mass spectrum of “water-soluble fraction” instead of  “peptide fraction”. As mentioned above, it is not demonstrated that all the components in this fraction are peptides.

Response: - As suggested by the reviewer in Figure 4 B) we have corrected the results analysis of LC/MS, yes it can be written as a ‘water-soluble fraction’ instead of a peptide-like fraction in our revised manuscript.

Round 2

Reviewer 1 Report

The study is relevant, but the peptide identification is crucial for its objectives. The authors did not present the chemical structure of any venom peptides. For this reason, I suggest to revise the discussion and conclusion sections. Please include the limitations of this study in the discussion section. 

Author Response

Reviewer #1 & suggestion to the author:

The study is relevant, but peptide identification is crucial for its objectives. The authors did not present the chemical structure of any venom peptides. For this reason, I suggest revising the discussion and conclusion sections. Please include the limitations of this study in the discussion section.

Response:

Thanks for all the critical suggestions. As suggested, we have revised the discussion and conclusion sections.
